# Evaluation of Delayed-Type Hypersensitivity to Antineoplastic Drugs—An Overview

**DOI:** 10.3390/cancers15041208

**Published:** 2023-02-14

**Authors:** Inés Roger, Paula Montero, Martín Pérez-Leal, Javier Milara, Julio Cortijo

**Affiliations:** 1Biomedical Research Networking Centre on Respiratory Diseases (CIBERES), Health Institute Carlos III, 28029 Madrid, Spain; 2Department of Pharmacology, Faculty of Medicine, University of Valencia, 46010 Valencia, Spain; 3Faculty of Health Sciences, Universidad Europea de Valencia, 46010 Valencia, Spain; 4Pharmacy Unit, University General Hospital Consortium, 46014 Valencia, Spain; 5Research and Teaching Unit, University General Hospital Consortium, 46014 Valencia, Spain

**Keywords:** antineoplastic, delayed-type hypersensitivity, preclinical test

## Abstract

**Simple Summary:**

The clinical practice nowadays encounters the problem of delayed-type hypersensitivity (DTH). DHT reactions are common with antineoplastic treatments, resulting in worsening patient quality of life. The range of symptoms in DHT reactions can vary from mild, such as self-limiting maculopapular eruptions, to severe, such as Stevens–Johnson Syndrome. In addition to limiting patients’ quality of life, these reactions also lead to economic losses due to withdrawal of affected drugs from the market and high hospitalization costs. Even so, there is no standard in vitro or in vivo method to evaluate the sensitizing potential of drugs in preclinical studies. This review is aimed at giving a comprehensive evaluation of in vitro and in vivo methods to detect DTH and possibly test antineoplastic hypersensitivity reactions caused by different antineoplastic families.

**Abstract:**

Nowadays, clinical practice encounters the problem of delayed-type hypersensitivity (DTH) induced by several drugs. Antineoplastic treatments are among the drugs which show an elevated proportion of DHT reactions, leading to the worsening of patients’ quality of life. The range of symptoms in DHT reactions can vary from mild, such as self-limiting maculopapular eruptions, to severe, such as Stevens–Johnson Syndrome. The development of these reactions supposes a negative impact, not only by limiting patients’ quality of life, but also leading to economic loss due to market withdrawal of the affected drugs and high hospitalization costs. However, despite this problem, there are no available standard in vitro or in vivo methods that allow for the evaluation of the sensitizing potential of drugs in the preclinical phase. Therefore, the aim of this review is to summarize the skin reactions caused by the different antineoplastic families, followed by a comprehensive evaluation of the in vitro and in vivo methods used to detect DTHs and that could be suitable to test antineoplastic hypersensitivity reactions.

## 1. Introduction

The World Allergy Organization defines hypersensitivity reactions as the symptoms caused by exposure to specific stimulus at doses that are usually well tolerated by normal individuals. These reactions can be allergic or non-allergic and are difficult to predict [1]. The different hypersensitivity reactions are classified according to its mechanism of tissue injury. The classification was proposed in the 1960s by Coombs and Gell [2] and can be found in Table 1. Immediate hypersensitivity (type I) is mediated by IgE immunoglobulins specific for allergens. Cytotoxic hypersensitivity reactions (type II) are characterized by the involvement of IgG and IgM antibodies that bind against the host’s self-antigens, causing extensive damage. In the third type (type III), IgG and IgM antibodies bind to antigens to form immune complexes that will activate the complement system, causing organ damage. Finally, the type IV reactions are denominated delayed type (DHT) and are T-cell mediated. T cells can then produce a direct damage or can activate other immune cells that drive tissue injury [3].

It is of special interests that systemically administered drugs can cause DTH reactions that cannot be predicted in the standard toxicity studies. These reactions are often in clinical trials or case reports, and therefore the knowledge of their pathophysiology is very poor [4]. The variety of the reactions is wide, as it can manifest as self-limiting eruptions to life-threatening reactions [5,6,7,8]. These reactions have become a safety concern given that the incidence of mild-to-moderate reactions may be underestimated [4,9]. In the oncology community, hypersensitivity reactions to antineoplastic drugs are usually unexpected given the different symptoms to the common toxicity of antineoplastics. DHT reactions to antineoplastic have been found in different mechanism-of-action groups, such as taxanes, platinum-containing compounds, epipodophyllotoxins, asparaginase, procarbazine, doxorubicin, and 6-mercaptopurine [10].

The DHT reaction develops in two phases: induction and elicitation. In the induction phase, allergen-specific T cells are generated in a process that usually does not produce clinical symptoms. The second phase is the activation or elicitation and occurs when a previously sensitized individual is re-exposed to the inducing allergen. In this stage, an exaggerated immune reaction to the sensitizing agent occurs. The allergen is recognized by the antigen-presenting cells (APCs), and keratinocyte and Langerhan cells release cytokines that attract effector T lymphocytes (mostly CD8 cytotoxic T lymphocytes). This response triggers apoptosis and the recruitment of inflammatory cells such as macrophages and granulocytes, which are responsible for the clinical skin symptoms such as erythema, swelling, and pruritus. Therefore, type IV hypersensitivity is considered to be delayed because it does not occur at the first contact with the substance, but in subsequent contacts. The symptoms usually develop within 2–14 days after the exposure and can appear on any area of the body surface regardless of where the first contact occurred. Generally, the mild reactions resolve 8 to 10 days after the discontinuation of the treatment, while the severe reactions require weeks to resolve [11].

However, all drug reactions do not fit in the Gell and Coombs’s classification. This is due to the heterogeneity in hypersensitivity reactions to drugs and to the possibility that more than one mechanism can develop simultaneously in a given reaction [8]. Therefore, DHT reactions are also difficult to describe and classify, because the mechanisms that drive its cytotoxicity are yet to be unraveled, and, usually, the information reported in the literature is limited [4].

## 2. Mechanisms of Delayed-Type Hypersensitivity

As DTH reactions can be driven by different mechanisms, DTH has been further subdivided into four types (IVa, IVb, IVc, and IVd) depending on the cytokines induced. All the subtypes are characterized by elicitation by antigen-presenting cells (APCs) or direct T cells; however, in each subtype, a different effector cell appears to be predominant.

In type IVa, T-helper 1 (Th1) cells activate macrophages, and the secreted cytokines include IFN gamma and TNF alpha. CD8 T-cell responses also take part in this subtype, as well as monocytes and macrophages. In type IVb, the response elicited is mediated by Th2 cells, and IL-4, IL-5, and IL-13 are the principal cytokines involved. Eosinophilic responses are also induced in these reactions. In the IVc subtype, the predominant effector cells are cytotoxic T cells, and therefore common cytokines include perforin, granzyme B, and granulosyn. Finally, in the IVd subtype, T-cells are involved with the characteristic cytokines IL8, GM-CSF, IL-5, and granulysin. In these reactions, other effector cells, such as neutrophils, can also be found [1,12].

According to the clinical manifestations of DHT, the reactions are classified into two groups: mild-to-moderate reactions and severe or life-threatening reactions. Mild to moderate reactions include different rash manifestations, such as macular/maculopapular rash eruptions, and dermatitis reactions [4,11,13]. Contrarily, severe reactions have a lower incidence, but they can be life threatening [10]. Amongst these reactions, erythema multiforme (EM)-like eruptions can be found, as well as Stevens–Johnson syndrome (SJS)/toxic epidermal necrolysis (TEN), drug rash with eosinophilia and systemic clinical manifestations (DRESS), and acute febrile neutrophilic dermatosis. DRESS appears as a manifestation of the type IVb reactions [13], SJS/TEN, usually the most severe, is included inside type IVc, while acute febrile neutrophilic dermatosis is included in type IVd.

There is no well-established diagnosis for the detection of DRESS [14,15,16]. Some of the characteristic clinical manifestations of DRESS include the development of maculopapular rashes within 3 weeks of initiating treatment, fevers greater than 38 degrees, lymphadenopathy, hematologic abnormalities, and facial edema. A skin biopsy typically reveals a perivascular lymphocytic infiltrate in the papillary dermis, with eosinophils, atypical lymphocytes and occasional spongiosis, and intercellular oedema in the epidermis clinically manifested by intraepidermal vesicles. Despite the clinical manifestations described, there is no consolidated guideline to guide DRESS therapy, particularly in the field of chemotherapeutic agents. [17]. The mortality rate has been estimated at about 10% [18].

Regarding SJS/TEN, this is characterized by purpuric rashes with blisters, atypical target lesions, cutaneous sloughing, and skin detachment [19]. Mortality rates in the general population range from 1% to 5% for SJS and 25% to 35% for TEN [20]. Due to the high mortality and morbidity rates associated with SJS/TEN, clinical recognition and early diagnosis are crucial. The diagnosis of this disease is usually based on histological findings; mucous membrane and skin manifestations [21]; and systemic manifestations, which include renal failure, respiratory distress, and high fevers. [22,23,24].

It is also possible for type IV to include acute reactions such as febrile neutrophilic dermatosis, which causes painful erythematous plaques or nodules [25]. Febrile neutrophilic dermatosis is commonly referred to as Sweet Syndrome (SS). This syndrome is an uncommon inflammatory disorder with multiple autoimmune, infectious, neoplastic, and pharmaceutical associations [25,26]. The pathology of drug-induced SS is characterized by sterile neutrophilic inflammation mediated by T cells [12,26]. Clinically, patients with this syndrome present with painful erythematous plaques or nodules, as well as pustular, bullous, and necrotizing variants. In addition to the cutaneous manifestations, there is also a disease pattern characterized by high fevers and multiorgan disease such as ocular, hepatic, pulmonary, neurological, renal, and cardiac involvement [25,27].

## 3. Delayed-Type Hypersensitivity Response to Antineoplastic

Cutaneous effects associated with antineoplastic drugs are quite frequent. In this review, we focus specifically on DTH. One of the handicaps of DTH reactions due to antineoplastic agents is that signs and symptoms are unpredictable and do not match known drug toxicity [10]. DTH reactions are frequently associated with certain categories of antineoplastic agents, such as taxanes and platinum-containing compounds [4]. However, throughout this review, DTH reactions associated with other antineoplastic families will be detailed.

Table 2 summarizes and reviews the DTH reactions associated with the main antineoplastic agents used today, organized by mechanism of action.

### 3.1. Alkylating Agents

#### 3.1.1. Metal Salts

Carboplatin, cisplatin, and oxaliplatin are used in the treatment of a variety of cancers, such as testicular, ovarian, lung, breast, cervical, stomach, prostate, colon, and rectal cancer [10]. Because of the vague or inconsistent terminology used to describe these reactions, it is unknown what percentage of patients treated with cisplatin or carboplatin experience delayed-type hypersensitivity reactions [9,112,113]. The most commonly observed symptoms related to DTH were rash and hands/palmar itching [35].

#### 3.1.2. Mustard Gas Derivates

Chlorambucil is primarily used for the treatment of several chronic lymphoproliferative diseases. Cutaneous adverse drug reactions are rarely reported; however, there are some described in the literature. On one hand, some cases describe chlorambucil-induced DRESS syndrome. A physical examination revealed widespread erythematous maculopapular confluent eruption, sparing the face. When chlorambucil was stopped, the eruption resolved with 2 weeks [18,28]. A patch test confirmed a positive reaction to chlorambucil [28]. On the other hand, there are some reports that detail chlorambucil-induced TEN [29,30,31]. Patients developed a confluent maculopapular erythema and large flaccid bullae on the trunk, legs, feet, and mucous membranes, with a fever up to 38 degrees. A skin-patch test and cutaneous biopsy were used to confirm the diagnosis in some patients [29]. Finally, other DTHs with variable clinical manifestations have been observed in patients administered with chlorambucil. Among them, we can highlight maculo-papular eruption and erythroderma [32,33,34].

Although there are other subclasses of alkylating agents, only those with associated DHT have been described.

### 3.2. Antitumor Antibiotics

#### 3.2.1. Anthracyclines

Doxorubicin is widely used in the treatment of carcinomas, sarcomas, and hematological cancers. Although numerous cases of skin effects have been reported in patients treated with doxorubicin, most of them are not of delayed-type hypersensitivity. A single case of TEN has been reported [36].

#### 3.2.2. Non-Anthracyclines

Bleomycin is frequently used as a chemotherapeutic agent to treat various types of malignant tumors. The cytotoxic effects of bleomycin cause several adverse reactions, especially in the lungs and skin. Different clinical cases [37,38] describe that bleomycin causes severe cutaneous adverse reactions, such as SJS. Additionally, there is an unusual report of a case of SJS that was induced by peplomycin, a bleomycin analogue [114].

### 3.3. Spindle Inhibitors

#### 3.3.1. Taxanes

Paclitaxel is used for the treatment of breast, ovarian, lung, bladder, prostate, melanoma, and esophageal cancer, as well as other types of solid tumor cancers. On the other hand, a number of cancer types have been approved for treatment with docetaxel, including breast, non-small cell lung, advanced stomach, head and neck, and metastatic prostate. The literature details that both antineoplastics generate DTH, causing severe reactions [13,19]. Several patients have reported chemotherapy-induced SJS/TEN as a side effect of docetaxel and, to a lesser extent, paclitaxel [39,40,41,42,43].

#### 3.3.2. Topoisomerase Inhibitors

Topoisomerases I and II are critical for DNA function and cell survival, and different studies have identified these enzymes as cellular targets for several clinically active anticancer drugs. Etoposide is semisynthetic analogue of podophyllotoxin that is used in the therapy of several forms of solid tumors, lymphoma and leukemia, usually in combination with other agents [115]. Topotecan is reported to be active against various carcinomas, namely cervical cancer, small cell lung cancer, breast cancer, and ovarian cancer. In relation to the delayed hypersensitization effects generated by these antineoplastics, a single case report has been published describing the association of etoposide with SJS [44], and topotecan treatment has induced one case of SS, according the literature [45].

### 3.4. Signal Transduction Inhibitors

Tyrosine kinase inhibitors have emerged as a new frontier of cancer therapy. These agents include inhibitor of BCR-ABL, epidermal growth factor receptor (EGFR), platelet-derived growth factor (PDGFR), c-Kit, vascular endothelial growth factor (VEGF), and human epidermal growth factor receptor 2 (HER2). In parallel with the evolving applications of these receptors, cutaneous toxicities associated with these agents have become more well-known [116]. In this section, we provide an overview of DTH reactions related to tyrosine kinase inhibitors.

#### 3.4.1. Multikinase

Imatinib is an inhibitor of the BCR-ABL kinase and is the standard first-line therapy for patients with chronic myeloid leukemia in chronic phase and in patients with locally advanced or metastatic gastrointestinal stromal tumors. Dasatinib is a second-generation BCR-ABL kinase inhibitor that also inhibits Scr family kinase, c-Kit, and PDGFR and has been approved as a second-line treatment for patients with myeloid leukemia after treatment failure with imatinib. Nilotinib, a novel oral aminopryrimidine derivate, does not affect Scr family kinases, but it can inhibit BCR-ABL, PDGFR, and c-Kit.

Imatinib is associated with severe adverse reactions in 5% of patients [47]. The most commonly reported responses are maculopapular eruptions, which occur in 66.7% of patients—although not all cases are severe [46]—and periorbital edema. On the other hand, the less common ones include SJS [47,48,49,50,51,117,118,119,120], TEN [58], sweet syndrome [25], and DRESS [52,53,54,55,56,57]. Of note, the discontinuation of imatinib treatment induced a clinical improvement. There have been fewer side effects reported with dasatinib and nilotinib. It is likely that the drugs are more effective and specific against BCR-ABL, as well as having a reduced affinity for c-Kit and PDGFR. Additionally, because the drugs are relatively limited in availability, there may be incomplete data [48]. Cases of SS caused by nilotinib [59] and dasatinib [25] have also been reported in the literature.

#### 3.4.2. VEGF

Regorafenib, sorafenib, and sunitinib are small-molecule inhibitors of the tyrosine kinase coupled to the VEGFR, as well as bevacizumab, which is a monoclonal antibody against VEGFA. All of these agents usually trigger dermatologic adverse events. Bevacizumab was approved for the treatment of metastatic colorectal cancer. In addition, other inhibitors of the VEGF signaling pathway have been developed and are currently approved for the treatment of various types of cancer. Sunitinib and sorafenib have improved the treatment of advanced renal cell carcinoma and gastrointestinal stromal tumor. Skin toxicity is one of the frequent adverse effects of these drugs, and skin rash has been reported in ~40% and ~20% for sorafenib and sunitinib, respectively [121]. Currently, SJS induced by sunitinib (n = 1) [60], sorafenib (n = 3) [62,63,64,65,66], and regorafenib (n = 1) [72] has been reported, although this syndrome is not very common in patients treated with these drugs. Interestingly, erythema multiforme induced by sorafenib was around 19–25% in the Japanese population, which is much higher than the Caucasian population [19,67,68,69,70]. However, although a genetic role in the adversity of the pathogenesis of drug reaction is possible, the different incidence of cutaneous adverse reactions among different ethnicities needs to be further investigated. Finally, maculopapular reactions induced by sunitinib [61] and sorafenib have also been described [71].

Regarding bevacizumab, the literature contains few reports describing type-IV delayed hypersensitivity reactions. The most frequent clinical manifestations are rashes and progressive exacerbations [73].

#### 3.4.3. EGFR

Epidermal growth factor receptor (EGFR) inhibitors are a new class of drugs for the treatment of various malignancies. There are currently four approved drugs: gefitinib, afatinib, and erlotinib for non-small cell lung carcinoma [122]; and cetuximab for colon cancer. Gefitinib and erlotinib are two low-molecular-weight anilinoquinazolines that penetrate through the cell membrane and block the tyrosine kinase to which EGFR is coupled. Afatinib is a dual kinase inhibitor of EGFR and HER2. Cetuximab is a chimeric IgG1 monoclonal antibody that blocks EGFR in its extracellular domain.

The safety profile of EGFR inhibitors is characterized by a cutaneous effect; among them is DTH. The most common group of drugs to induce SJS/TEN (n = 9) is0 EGFR inhibitors [74,75,76,77,78,79,80,81,82]. Researchers hypothesize that SJS/TEN caused by EGFR inhibitors results in extensive erosion of the epidermis due to epidermal differentiation and re-epithelialization because of the irreversible inhibition of EGFR [76]. However, there are not many works in the literature describing the involvement of these drugs in other skin conditions related to delayed-type hypersensitivity.

#### 3.4.4. BRAF and MEK1/2

BRAF is an upstream activator of the MAPK pathway, which is involved in cell differentiation, migration, and proliferation [123]. BRAF is mutated in 40–60% of melanomas [124]. BRAF inhibitors include vemurafenib, dabrafenib, and encorafenib. On the other hand, MEK1 and -2 are dual-specificity kinases that activate ERK1/2 by phosphorylating [125]. Trametinib, cobimetinib, and binimetinib are targeting the MEK1 and -2 kinases and are currently approved in combination with BRAF inhibitors. DTH effects have only been reported in patients treated with vemurafenib, dabrafenib, tratetinib, and cobimetinib. BRAF inhibitors have been associated with eight cases of SJS/TEN [83,84,85,86,87,88,89,90]. It has been reported that vemurafenib and cobimetinib were associated with a case of SJS [83]. The remaining seven cases occurred in the setting of vemurafenib alone [84,85,86,87,88,89,90]. Moreover, BRAF inhibitors have also been reported to cause DRESS [83,91,92,93,94,95]. A total of 7171 reports for DRESS have been reported in the FDA database, of which 125 were associated with vemurafenib [96]. In conclusion, we can say that, within this group of drugs, vemurafenib was the drug that, with higher probabilities, triggered severe delayed-type cutaneous reactions. Finally, cases of SS have also been reported in patients treated with dabrafenib, trametinib [99], and vemurafenib [97,98].

It is important to note that, although we have mainly detailed the more severe clinical manifestations, such as SJS/TN or DRESS, mild manifestations also occur. However, the vague or incoherent terminology used to describe these reactions causes the type of hypersensitivity that generates the clinical manifestation not to be specified.

### 3.5. Antimetabolites

#### 3.5.1. Purine Analogs

The purine analogues are antimetabolites that interfere with nucleic acid synthesis. Cladribine and fludarabine are nucleoside analogs that are active in the systemic treatment of a variety of lymphoproliferative diseases. Only one case of SJS associated with cladribine [100] and one with fludarabine [101] have been reported. In addition, one case of cladribine-associated with TEN has also been reported [126].

#### 3.5.2. Pyrimidine Analogs

The pyrimidine analogues have wide applications in the management of both hematological and solid cancer and include cytarabine, gemcitabine, and azacytidine. Hypersensitivity reactions have been described in many patients; however, DTH is rare. Three cases of SJS/TEN were described after gemcitabine administration [104,105,106], one case of SJS after capecitabine administration [102] and two cases of TEN after cytarabine administration [108,109]. The diagnosis was confirmed by a skin biopsy of an affected area. Moreover, these drugs are also related to SS, specifically capecitabine [103], gemcitabine [107], and azacytidine [110,111].

## 4. Preclinical Test

Despite the serious adverse reactions linked to DTH, at present, there are no validated in vivo or in vitro methods for assessing the sensitizing potential of a drug during the preclinical phase. The following section provides an overview of in vivo and in vitro methods.

### 4.1. In Vivo Test for the Preclinical Assessment

The traditional methods for detecting sensitization potential in humans utilize the guinea pig test. This test has been shown to perform well in predicting skin sensitization caused by environmental or industrial chemicals [127]. However, while it has been used for the evaluation of pharmaceuticals since the 1980s, there are not much data to support its ability to identify systemic drug hypersensitivity. The mouse Local Lymph Node Assay (LLNA) has been put forward as an alternative assay to the guinea pig assay for predicting skin sensitization and has been extensively evaluated by ICCVAM (The Interagency Coordinating Committee on the Validation of Alternative Methods) in recent years [128,129]. Since the LLNA method utilizes topical exposure test substances, resulting in variable levels of local and systemic exposure, the method was modified to administer the drug by subcutaneous injection of test compounds to achieve a known systemic exposure to the drug. The LLNA modification is called the Local Lymph Node Proliferation Assay (LNPA), and the data suggest that this method might be useful to accurately predict the DTH reaction to systemically administered pharmaceuticals [130,131]. For both LLNA and LLNP, mice must receive a test substance once daily for three straight days and then 2 days of rest before receiving 3H-thymidine intravenously. Five hours after receiving thymidine injections, the animals are sacrificed, and draining lymph nodes are examined by scintillation counting. The incorporation of 3H-thymidine into the DNA of lymph node-resident leukocytes serves as a marker for T-cell activation related to allergic sensitization [131]. The DTH response can be evaluated by assessments of localized swelling, leukocyte infiltration of the challenged tissues, and Th1-associated cytokine profiling [132]. However, at present, none of these methods is available (or validated) from regulatory agencies.

### 4.2. In Vitro Test for the Preclinical Drug Assessment

The development of in vitro models to predict drug-mediated DTH is the result of international initiatives to reduce the use of research animals. An enhancement in safety and maybe a decreased probability of market removal would result from the development of in vitro assays to identify the sensitization potential during the drug development period [133].

Since conventional medications have low molecular weights, T cells cannot detect them. As a result, in order for low-molecular-weight drugs to be detected by T cells, they must first bind to a protein [134,135]. In addition, drugs may alter the antigen presentation process or may cause cellular damage and release self-antigens (e.g., DNA or histones), for which there is no tolerance [136], leading to the development of hypersensitivity reactions.

The majority of the in vitro methods that have been proposed profits from a rationalistic approach based on the notion that allergenic medicines and chemical allergens share similar processes of cell activation. To identify medicines that may have the potential to elicit in vivo hypersensitive reactions, this assay can be simply introduced into drug development [137].

Direct Peptide Reactivity Assay (DPRA) for assessing haptenization [138] and methods based on dendritic cells [131,138,139] are potentially useful and relevant for systemically administered drugs [137,140]. Table 3 summarizes the in vitro preclinical assays described for the study of sensitizing substances.

### 4.3. The T-Cell Priming Assay

For the purpose of identifying contact allergens which induce T-lymphocyte responses, the T-cell priming assay was developed. The assay uses human dendritic cells, a defined naïve T-cell population, and optimized cell culture conditions for antigen-specific activation and the expansion of naïve T cells. Thus, the assay allows the identification of reactive CD4+ and CD8+ T cells as early as 10 days after their primary activation.

This multiparametric flow-cytometry-based assay identifies the cytokines IFN-γ and TNF-α production in a subset of CD4+ and CD8+ T-cells populations [141,142]. To test this, naïve T cells and monocyte-derived dendritic cells (MoDCs) pulsed with the test chemical were co-cultured in the presence of feeder cells, costimulatory CD28 antibody, and cytokines. An analysis of the chemical-specific T-cell response can be a useful in vitro assay for hazard identification in immunotoxicology since it is the immune system’s most focused response to allergens [141]. This assay may be a valuable, highly specific element of an integrated testing strategy for the identification of chemicals and drugs that cause T-cell-mediated reactions.

### 4.4. Myeloid U937 Skin Sensitization Test (MUSST)

A proposed in vitro approach to measure skin sensitivity is the modified Myeloid U937 Skin Sensitization Test (MUSST). This assay addresses the third key event, namely dendritic cell activation by quantifying changes in the expression of cell surface markers following exposure to the drug. The idea behind this approach is that in order to generate T-cell memory, which is necessary for allergy sensitization, antigen-presenting cells such as dendritic cells must be activated [131]. The U937 human myeloid cell line was used to represent dendritic cell activation following exposure to sensitizers by evaluating the induction of CD86 expression by flow cytometry after 48 h of chemical treatment. When CD86 induction exceeds the cutoff of 1.5-fold with respect to vehicle-treated cells at any tested concentration, showing a cell viability of less than 70% in at least two independent experiments, the test substance is predicted to have a dendritic-cell-activating potential that is indicative of being a sensitizer [137,143]. This test has been validated with 65 compounds and compared with human and/or LLNA data [143]. The technical limitations and limitations with regard to predictivity are that highly cytotoxic chemicals or chemicals that interfere with the detection systems cannot always be reliably tested. Moreover, due to the aqueous nature of the cell medium, solubility issues can occur when testing lipophilic substances.

### 4.5. The Human Cell Line Activation Test (h-CLAT)

The principle of h-CLAT is very similar to that of MUSST, but in this case, h-CLAT quantifies changes in the human cell line THP-1 in the expression of both CD86 and CD54 following 24 h exposure to the test chemical. THP-1 cells are monocyte-derived cells that have been shown to produce dendritic-cell-like responses following exposure to skin-sensitizing drug. As in MUSST, changes in surface marker expression are measured by flow cytometry. In addition, to study whether the overexpression of surface markers occurs at subcytotoxic concentrations, a measurement of cytotoxicity is performed [137,143,144]. An h-CLAT prediction is considered positive if the relative fluorescence intensity (RFI) of CD86 is equal to or greater than 150% at any tested dose (with cell viability ≥ 50%) or if the RFI of Cd54 is equal to or greater than 200% at any tested dose (with cell viability ≥ 50%). Regarding the validation status of the method, in this case, the assay was evaluated in a reliability validation study (EURL ECVAM, 2015) and it is an officially adopted test method (OECD TG 442E). The results generated in the validation study and published studies [144] indicate that the accuracy of the h-CLAT in discriminating sensitizers from non-sensitizers is 85%, with a specificity of 66% and a sensitivity of 93% when compared to LLNA results.

### 4.6. IL-8 Luc Assay

As mentioned above, the activation of dendritic cells is one of the key steps in skin sensitization. This step is characterized by an overexpression of CD45 and CD86. In addition, gene and protein overexpression of interleukin-8 (IL-8) have also been observed. IL-8 expression is associated with the activation of dendritic cells (key event 3). Therefore, IL-8 expression has been suggested as another biomarker to identify sensitizers in monocyte-derived dendritic cells. In IL-8 Luc assay, IL-8 expression is measured in the THP-1-derived IL-8 reporter cell line (THP-G8) as luciferase activity [145]. The assay has some advantages in terms of technical performance and time required compared to existing methods concerning key event 3. The validation dataset showed an accuracy of 82.4%, a sensitivity of 79.2%, and a specificity of 90% [146].

### 4.7. The THP-1 Activation Assay

Using THP-1 activation, low-molecular-weight chemicals can be evaluated for their allergenic potential in vitro based on the molecular mechanisms underlying skin and systemic sensitization. In order to take advantage of the human THP-1 cell line’s capacity to recognize contact and respiratory allergens, it was produced during the European project SENS-IT-IV that provided information on key event 3. The technique was based on the assessment of IL-8 production and CD86 and CD54 expressions for the purpose of identifying allergenic substances [147,148]. A chemotactic peptide for neutrophils and T cells, IL-8, has been demonstrated to be a valuable biomarker to distinguish sensitizers from non-sensitizers [145]. The THP-1 activation assay comprises the study of CD86 expression alone and/or in combination with CD54 expression for the identification of chemical sensitizers in addition to the investigation of IL-8 production. The h-CLAT approach additionally investigates CD54 and CD86 as parameters [149].

In addition to the studies described above, recent evidence indicates that oxidative stress is involved in DTH reactions [150].

One of the catalysts for DC maturation appears to be the chemically driven oxidation of cell-surface thiols, as this causes an intracellular redox imbalance and a stress-related signal. The Keap1/Nrf2 signaling pathway is responsible for detecting electrophilic stress in cells and causing the activation of genes involved in reactive chemical species’ defense or neutralization [151] Chemical sensitizers have been found to cause oxidative stress in human monocyte-derived dendritic cells, as was detected by the glutathione GSH/GSSG ratio as a redox marker. It is possible that the electrophilic characteristics of chemical sensitizers may be regarded by DCs as a danger signal, resulting in DC maturation, as evidenced by the fact that the reduction of the glutathione GSH/GSSG ratio was followed with the upregulation of CD86 and activation of p38 MAPK [152,153].

These in vitro assays have been used to test more than 145 substances, but in addition, dendritic-cell-based assays have been used to test some drug sensitizers, such as benzocaine, hydroquinone, p-benzoquinone, and diphenylcyclopropenone which were tested and classified correctly [137,143,147,154].

In addition to assays based on dendritic cell activation, there are other assays that report other key events in skin sensitization, such as direct peptide reactivity assay (DPRA) and KeratinoSens.

### 4.8. Direct Peptide Reactivity Assay (DPRA)

DPRA is a cell-free chemical assay that evaluates the reactivity of test compounds by using peptides containing lysine and cysteine. In the molecular initiating event, reactive electrophiles form bonds with cysteine(s) or lysine(s). Activating the innate immune system through haptens also triggers the activation of T-cells. Therefore, this assay is deemed to address the molecular initiating event of the skin-sensitization adverse outcome pathways (AOPs), namely protein reactivity, and quantifies the reactivity of test chemicals toward synthetic peptides containing either lysine or cysteine [138]. Compared to human data, 84% of non-sensitizers and sensitizers yielded consistent results in the DPRA [155]. As far as technical limitations are concerned and limitations with regard to predictivity, it should be noted that the method is not suitable for testing highly hydrophobic chemicals, that test chemicals with the same retention time as the cysteine and the lysine peptides provide inconclusive results, and that the test chemicals that are required to be metabolically activated to act as sensitizers (pro-haptens) cannot be detected as being reactive in the DPRA.

### 4.9. KeratinoSens

Human keratinocyte cell lines are used in the manufacture of KeratinoSens assays, and the activation of the Nrf2/Keap1/ARE signaling pathway is being studied. For sensing the specific activation of Nrf2, which is a key regulator of the keratinocyte inflammatory response, it uses an antioxidant response element (ARE)-coupled luciferase assay [156]. The second key event of the skin-sensitization AOP, namely keratinocytes’ activation, evaluates the Nrf2-mediated activation of antioxidant response element (ARE)-dependent genes (OECD, 442D) through luciferase induction. ARE-regulated genes have been reported to be induced by skin sensitizers [151,154,157]. As a result of covalent modification of Keap1’s (Kelch-like ECH-associated protein 1) cysteine residue by small electrophilic substances such as skin sensitizers, the sensor protein Keap1 is able to dissociate from Nrf2 (nuclear factor-erythroid 2-related factor 2). The dissociated Nrf2 can then activate ARE-dependent genes such as those coding for phase II detoxifying enzymes [158,159]. This methos has been evaluated in a validation study for reliability (EURL ECVAM, 2014) and by an officially adopted test method (OECD TG 442D). The accuracy of the KeratinoSens in discriminating sensitizers from non-sensitizers is 77% (n = 201), with a sensitivity of 78% and a specificity of 76% when compared to LLNA results [160].

To study the sensitizing potential of some drugs, these assays have been performed individually [131,133,149] or in combination [161]. However, some studies suggest that the integration of several techniques involving key mechanistic processes may result in a more accurate prediction [161]. Therefore, these techniques could be used for the rapid screening of multiple drugs that cause DTH.

## 5. Preclinical Assays to Detect Antineoplastic-Mediated-DTH

As mentioned above, these techniques offer, with quite high sensitivity, a great potential to determine the sensitizing capacity of the drug. However, although these techniques have been used to study the DHT-developing potential of many other drugs [130,131,149,162,163] or medical devices [131,164], few antineoplastic drugs have been tested with the preclinical assays described above. Table 4 summarizes the antineoplastic drugs tested with the in vitro or in vivo preclinical techniques detailed above.

The guinea pig model was formerly used to perform hypersensitivity studies. This model was replaced by the LLNA or LNPA model. However, some drugs have been tested for their sensitizing capacity by using this method. The DTH-generating potential of bleomycin was studied in guinea pigs, and it was reported that bleomycin administration produced a significant increase in skin reactivity [165].

Due to the intention to reduce animal experimental models to a minimum, the sensitizing capacity of some of the antineoplastic drugs listed above has been tested by in vitro assays. H-CLAT, DPRA, and KeratinoSens are some of the most commonly used in vitro tests for the study of the sensitizing potential of drugs, as they describe the key processes occurring in the drug sensitization process. Although evidence suggests that these preclinical assays are capable of determining the sensitizing potential of antineoplastic drugs with high reliability, few studies have been performed with antineoplastic drugs, and the study should be expanded.

## 6. Conclusions

It is well-known that DHT reactions are rare [166], but in any case, they usually cause a lot of discomfort for patients and can become dangerous, especially in severe cases such as anaphylaxis, SJS, TEN, or DRESS. In addition, they are a burden on healthcare systems, as severe clinical pictures result in longer stays. On the other hand, the withdrawal of a drug from the market is also an important economic issue due to the very high costs associated with drug development [136]. The development of alternative in vitro assays to detect sensitization potential during the development phase of a drug would increase safety and possibly reduce the risk of market withdrawal [133]. The integration of several in vitro assays, instead of using a single test, could have sufficient sensitivity to detect the potential of antineoplastic drugs to trigger DTH. Some studies have shown that the use of three techniques that study key mechanistic events, namely DPRA, hCLAT, and KeratinoSens, could be used as a rapid screening for drugs that cause DTH. However, today there are no validated in vivo or in vitro methods to assess the sensitization potential of a drug during the preclinical phase. Although there are many in vivo or in vitro tests that have succeeded in correctly identifying sensitizing drugs in this review, the need to test antineoplastics with these tests and validate the different assays has been highlighted.

## Figures and Tables

**Table 1 cancers-15-01208-t001:** Gell and Coombs Classification of hypersensitivity reactions.

Hypersensitivity Reaction	Type	Immune Mediators
Immediate	Type I	IgE mediated
Cytotoxic	Type II	IgG/IgM mediated
Immune complex	Type III	Immune complex mediated
Delayed type	Type Iva	Th1 cell-mediated macrophage activation
	Type IVb	Th2-cell-mediated eosinophilic inflammation
	Type IVc	Cytotoxic T cell mediated
	Type IVd	T-cell-mediated neutrophilic inflammation

Ig: immunoglobulin.

**Table 2 cancers-15-01208-t002:** Clinical manifestations of DTH related to antineoplastics drugs.

Mechanism of Action	Subgroup	Causative Drug	Clinical Manifestations of DTH	Reference
Alkylating agents	Mustard gas derivates	Chlorambucil	DRESS	[18,28]
			TEN	[29,30,31]
			Maculo-papular eruption and erythroderma	[32,33,34]
	Metal salts	Carboplatin	Rash and hands/palmar itching	[35]
Antitumor antibiotics	Anthracyclines	Doxorubicin	TEN	[36]
	Non-anthracyclines	Bleomycin	SJS	[37,38]
Spindle inhibitors	Taxanes	Docetaxel	SJS	[39,40,41,42]
		Paclitaxel	SJS	[43]
	Topoisomerases inhibitors	Etoposide	SJS	[44]
		Topotecan	SS	[45]
Signal transduction inhibitors	Multikinases	Imatinib	Maculopapular eruptions	[46]
			SJS	[47,48,49,50,51]
			SS	[25]
			DRESS	[52,53,54,55,56,57]
			TEN	[58]
		Nilotinib	SS	[59]
		Dasatinib	SS	[25]
	VEGF	Sunitinib	SJS	[60]
			Maculopapular eruptions	[61]
		Sorafenib	SJS	[62,63,64,65,66]
			Erythema multiforme	[19,67,68,69,70]
			Maculopapular eruptions	[71]
		Regorafenib	SJS	[72]
		Bevacizumab	Rash	[73]
	EGFR	Afatinib	SJS/TEN	[74,75]
		Erlotinib	SJS/TEN	[76]
		Gefitinib	SJS/TEN	[77,78]
		Cetuximab	SJS/TEN	[79,80,81,82]
	BRAF	Vemurafenib	SJS/TEN	[83,84,85,86,87,88,89,90]
			DRESS	[83,91,92,93,94,95,96]
			SS	[97,98]
		Dabrafenib	DRESS	[83]
			SS	[99]
	MEK1/2	Cobimetinib	SJS/TEN	[83]
		Trametinib	SS	[99]
Antimetabolites	Purine analogs	Cladribine	SJS/TEN	[100]
		Flurabine	SJS	[101]
	Pyrimidine analogs	Capecitabine	SJS	[102]
			SS	[103]
		Gemcitabine	SJS/TEN	[104,105,106]
			SS	[107]
		Cytarabne	SJSTEN	[108,109]
		Azacitidine	SS	[110,111]

Abbreviations: DRESS, drug rash with eosinophilia and systemic clinical manifestations; HER2, human epidermal growth factor receptor 2; EGFR, epidermal growth factor receptor, PDGFR, platelet-derived growth factor; SJS, Stevens–Johnson Syndrome; SS, Sweet Syndrome; TEN, toxic epidermal necrolysis; VEGF, vascular endothelial growth factor.

**Table 3 cancers-15-01208-t003:** In vitro preclinical assays described for the study of sensitizing substances.

Test	Cells	Marker	Assay
T-cell priming assay	Naïve T cells and MoDC	IFN-γ and TNF-α	Flow cytometry
h-CLAT	THP1	Cd54 and cd86 expression and viability	Flow cytometry
MUSST	U937	CD86	Flow cytometry
IL-8 Luc assay	THP-1-derived IL-8 reporter cell line	IL-8	Luminescence
THP-1 activation assay	THP1	IL-8, CD86 and CD54	Flow cytometry and ELISA
DPRA	cell-free	protein reactivity	HPLC
KeratinoSens	KeratinoSens^TM^ cell line (keratinocytes)	Viability and Nrf2 activation	MTT and luminescence

Abbreviations: DPRA, direct peptide reactivity assay; ELISA, enzyme-linked immunosorbent assay; h-CLAT, Human Cell Line Activation Test; HPLC, high-performance liquid chromatography; IL, interleukin; IFN, interferon; MoDC, monocyte-derived dendritic cells; MUSST, Myeloid U937 Skin Sensitization Test; TNF, tumor necrosis factor.

**Table 4 cancers-15-01208-t004:** Antineoplastic drugs tested with the in vitro or in vivo preclinical assays.

Type of Preclinical Test	Test	Drugs Tested
In vivo	Guinea pig model	Bleomycin [165]
In vitro	h-CLAT	Docetaxel [161], paclitaxel [161], imatinib [161], nilotinib [161], dasatinib [161], sunitinib [161], sorafenib [161], regorafenib [161], bevacizumab [161]
	DPRA	Docetaxel [161], paclitaxel [161], imatinib [161], nilotinib [161], dasatinib [161], sunitinib [161], sorafenib [161], regorafenib [161], bevacizumab [161]
	KeratinoSens	Docetaxel [161], paclitaxel [161], imatinib [161], nilotinib [161], dasatinib [161], sunitinib [161], sorafenib [161], regorafenib [161], bevacizumab [161]

Abbreviations: DPRA, Direct peptide reactivity assay; h-CLAT, Human Cell Line Activation Test.

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
