# Peer review of "Evaluation of Delayed-Type Hypersensitivity to Antineoplastic Drugs—An Overview"

_cancers, 2023, doi:10.3390/cancers15041208_

Round 1
Reviewer 1 Report
The review by Roger et al. addresses the DHT adverse events linked to the use of antineoplastic drugs. As a general comment, the subject is of clinical pertinence as it is covering an important health issue in the management of cancer patients’ care. The manuscript is however suffering from several issues that need to be addressed prior to potential publication.
- The review deserves editing as several mistakes can be noticed throughout the manuscript
- The hypersensitive reactions described in Table 1 could/should be complemented by the sub-classification of the grade IV events. It would indeed facilitate the reading for non-experts.
- Lane 113 : what is spongiosis, please describe
- Section 3.1, alkylating agents contains more than metal salts and mustard gas derivatives… There is no reference to other sub-classes of alkylating agents such as oxazaphosphorines (cyclophosphamide and derivatives) or Triazenes and hydrazines (temozolomide, dacarbazine) that are also routinely used in the clinic. Is it due to the fact that these alkylating agents are not associated with DHT or that they have been omitted? Please specify.
- In the same line, other classes of chemos are not mentioned at all, such as topoisomerase I poisons, or nucleotide analogues. Most importantly immune checkpoint inhibitors that are often used in combination with chemo and are associated with skin rash are also absent from the review…? This should be addressed.
- section 3.4.2, the authors should be more precise when frequency of the events is mentioned. Percentages instead of “not very frequent” should be mentioned to give a sense of what is “not frequent” for non-experts.
In the same section, no explanation is provided to explain the difference between Japanese and Caucasian populations. It would not be unexpected that a “polymorphic”-based rational may be involved in that difference. Is it the case?
- Section 3.4.4: trametinib, cobimetinib and binimetinib are NOT B-raf inhibitors but are targeting the MEK1/2 kinases instead and are currently approved in combination with RAF inhibitors…Please correct in the etxt and in the Table. In this line, is it known whether adding a MEK inhibitor to BRAF inhibitors is increasing the severity of the DHT? Please specify. Are the symptoms with MEK inhibitors the same than those observed for RAF inhibitors?
- Section 4.1 would beneficiate for some details as far as the tests are concerned. The principle of each test should be detailed a little bit more, especially for non experts. Also, details on the settings, on the cost, on the specificity, on the endpoint, etc…should be included.
- Table 3 should be moved closer to the section where it is mentioned first.
- Line 388, please specify AOP
- The conclusions are vague (probably because there is no robust test that is approved). The authors should at least give potential directions and highlight what could be the future of these tests: whether one will take over the others, which one could be the most reliable, the less expensive one, the one that could easily be adapted to the clinical routine?...
Author Response
Response to Decision Letter
The authors would like to express their gratitude for the work carried out by the referees in reviewing this manuscript. The revised manuscript has taken into account all the comments and criticisms raised by the reviewers thus improving, in our opinion, the quality of the revised version.
To ease the review process, in our reply, the reviewer’s comments are pasted just preceding the corresponding replies, as indicated. Also, the main changes introduced in the revised manuscript are highlighted in yellow.
- The hypersensitive reactions described in Table 1 could/should be complemented by the sub-classification of the grade IV events. It would indeed facilitate the reading for non-experts.
Reply
We agree with reviewer 1. We have complemented the table with the sub-classification of the grade IV events. All the corrections have been highlighted in yellow in the revised manuscript.
- Lane 113: what is spongiosis, please describe.
Reply
We have described briefly what is spongiosis. It has been highlighted in yellow in the revised manuscript
- Section 3.1, alkylating agents contains more than metal salts and mustard gas derivatives… There is no reference to other sub-classes of alkylating agents such as oxazaphosphorines (cyclophosphamide and derivatives) or Triazenes and hydrazines (temozolomide, dacarbazine) that are also routinely used in the clinic. Is it due to the fact that these alkylating agents are not associated with DHT or that they have been omitted? Please specify.
Reply
Effectively, although other alkylating agents exist, only those that generate DHT have been described. Alkylating agents such as cyclophosphamide, mechlorethamine or procarbazine are associated with hypersensibility type I related to the presence of drug-specific IgE resulting in mast cell and basophil activation (Krutchik AN, Buzdar AU, Tashima CK. Cyclophosphamide-induced urticaria. Occurrence in a patient with no cross-sensitivity to chlorambucil. Arch Intern Med 1978;138:1725-17; Lakin JD, Cahill RA. Generalized urticaria to cyclophosphamide: Type I hypersensitivity to an immunosuppressive agent. J Allergy Clin Immunol 1976;58:160-171; Millard LG, Rajah SM. Cutaneous reaction to chlorambucil. Arch Dermatol 1977;113:1298; Glovsky MM, Braunwald J, Opelz G, et al. Hypersensitivity to procarbazine associated with angioedema, urticaria, and low serum complement activity. J Allergy Clin Immunol 1976;57:134-140). To avoid confusion, we have added it to the main text.
- In the same line, other classes of chemos are not mentioned at all, such as topoisomerase I poisons, or nucleotide analogues. Most importantly immune checkpoint inhibitors that are often used in combination with chemo and are associated with skin rash are also absent from the review…? This should be addressed.
Reply
Reviewer 1 is right. We had neglected to mention these subtypes of antineoplastics as the reviewer points out, they produce type IV hypersensitivity reactions. They have therefore been added to the main text and in table 2. It has been highlighted in yellow in the revised manuscript.
- section 3.4.2, the authors should be more precise when frequency of the events is mentioned. Percentages instead of “not very frequent” should be mentioned to give a sense of what is “not frequent” for non-experts.
Reply
We agree with reviewer 1. To be more precise, the observed skin toxicity rates have been added. However, in the case of Steve-Jhonson syndrome, percentages could not be given because they are not known; only a few cases have been described in which treatment with these antineoplastics is associated with the development of the syndrome. In the absence of this, the number of cases described for each antineoplastic has been added to the text. It has been highlighted in yellow in the revised manuscript.
-In the same section, no explanation is provided to explain the difference between Japanese and Caucasian populations. It would not be unexpected that a “polymorphic” based rational may be involved in that difference. Is it the case?
Reply
In the article that indicates this difference, it mentions that this could imply a possible genetic role in the pathogenesis of adverse drug reactions. But, the different incidences of cutaneous adverse reactions among different ethnicities need to be further investigated. I have added this information to the manuscript for the avoidance of doubt. It has been highlighted in yellow in the revised manuscript.
- Section 3.4.4: trametinib, cobimetinib and binimetinib are NOT B-raf inhibitors but are targeting the MEK1/2 kinases instead and are currently approved in combination with RAF inhibitors…Please correct in the etxt and in the Table. In this line, is it known whether adding a MEK inhibitor to BRAF inhibitors is increasing the severity of the DHT? Please specify. Are the symptoms with MEK inhibitors the same than those observed for RAF inhibitors?
Reply
We agree with reviewer 1. We have corrected the text and the tablet and indicated that trametinib, cobimetinib and binietinib are not BRAF inhibitors but are targeting the MEK1/2 kinases. According to the literature, although cases of SJS have been described in both BRAF and MEK1/2 inhibitors, vemurafenib is the drug that triggers the most severe delayed-type cutaneous reactions. It has been highlighted in yellow in the revised manuscript.
- Section 4.1 would beneficiate for some details as far as the tests are concerned. The principle of each test should be detailed a little bit more, especially for non experts. Also, details on the settings, on the cost, on the specificity, on the endpoint, etc…should be included.
Reply
We agree with reviewer 2. We have expanded the description of the techniques including further details on the technique, as well as limitations and sensitivity. It has been highlighted in yellow in the revised manuscript.
- Table 3 should be moved closer to the section where it is mentioned first.
Reply
We agree with reviewer 1. We have moved the table so that it is presented right after it is mentioned in the text.
- Line 388, please specify AOP
Reply
We agree with reviewer 1. We have specified AOP in the main text.
- The conclusions are vague (probably because there is no robust test that is approved). The authors should at least give potential directions and highlight what could be the future of these tests: whether one will take over the others, which one could be the most reliable, the less expensive one, the one that could easily be adapted to the clinical routine?...
Reply
We agree with reviewer 1. Based on the literature we have added in the main text that the integration of several in vitro techniques offers a higher sensitivity to detect the ability of antineoplastic drugs to trigger DHT. Specifically, the use of three techniques that study key mechanistic events such as DPRA, hCLAT and Keratinosens could be used as a rapid screening for drugs that cause DTH. It has been highlighted in yellow in the revised manuscript.

Reviewer 2 Report
Dear authors,
I find delayed-type hypersensitivity (HS) induced by antineoplastic drugs is a delicate and at the same time important subject.
I read the manuscript several times to be sure that it is complete.
I believe the subchapter are well designed .
In Introduction I suggest to rephrase line 72/73 in order not to be mistaken with HS type 1.
In chapter 3 I suggest that Table 2 to be presented right after it is mentioned into the text.
Chapter 4 is also well organised but I still believe the Tables should be placed right after they are mentioned. The non validated methods for pre-clinical testing antineoplastic drugs for HS are well described.
I suggest to extend subchapters: The T cell Priming assay; Myeloid U937 Skin Sensitization Test (MUSST) and IL-8 Luc assay, some readers would like more details.
The conclusions are according to the whole manuscript.
I congratulate the authors and I recommend publication after minor corrections.
Author Response
Response to Decision Letter
The authors would like to express their gratitude for the work carried out by the referees in reviewing this manuscript. The revised manuscript has taken into account all the comments and criticisms raised by the reviewers thus improving, in our opinion, the quality of the revised version.
To ease the review process, in our reply, the reviewer’s comments are pasted just preceding the corresponding replies, as indicated. Also, the main changes introduced in the revised manuscript are highlighted in yellow.
Dear authors,
I find delayed-type hypersensitivity (HS) induced by antineoplastic drugs is a delicate and at the same time important subject.
I read the manuscript several times to be sure that it is complete.
I believe the subchapter are well designed .
In Introduction I suggest to rephrase line 72/73 in order not to be mistaken with HS type 1.
Reply
We agree with reviewer 2. We have rephrased line 72/73 to not confuse with HS type 1. All the corrections have been highlighted in yellow in the revised manuscript
In chapter 3 I suggest that Table 2 to be presented right after it is mentioned into the text.
Reply
We agree with reviewer 2. We have moved the table so that it is presented right after it is mentioned in the text.
Chapter 4 is also well organised but I still believe the Tables should be placed right after they are mentioned. The non validated methods for pre-clinical testing antineoplastic drugs for HS are well described.
Reply
The reviewer is right. As in the previous case, we have moved the table so that it appears just after being mentioned.
I suggest to extend subchapters: The T cell Priming assay; Myeloid U937 Skin Sensitization Test (MUSST) and IL-8 Luc assay, some readers would like more details.
Reply
We agree with reviewer 2. We have expanded the description of the above-mentioned techniques including further details on the technique, as well as limitations. It has been highlighted in yellow in the revised manuscript.
The conclusions are according to the whole manuscript.
I congratulate the authors and I recommend publication after minor corrections.

Round 2
Reviewer 1 Report
The manuscript has been drastically improved and has addressed all the points that were raised. The informations that are included in the Review are of clinical pertinence and the manuscript can be published in its present form.